# Controllable Preference Alignment for Ambiguous Medical Image Segmentation via Text and Dice Guidance

## Abstract

In medical imaging, different experts often provide different but equally valid segmentations, making ambiguity an inherent challenge. A good model should therefore capture this variability by producing a distribution of plausible masks, rather than a single deterministic output. Diffusion models are well-suited for this task because of their ability to generate diverse samples, but standard training does not guarantee clinically meaningful segmentations. Prior work in ambiguous segmentation, like CIMD (Rahman et al., 2023), uses diffusion but lacks semantic control. We present Text- and Dice-Guided Diffusion-DPO (TDG-DiffDPO), a novel multi-modal framework that makes diffusion-based segmentation both controllable and clinically aligned. Our method conditions the model on input images and descriptive text from clinical metadata, and adapts Direct Preference Optimization (DPO) (Rafailov et al., 2023) by using Dice-based signals from multi-rater annotations instead of subjective human feedback. We explore three preference strategies and find that a consensus-based Mean Dice signal is most effective. With DDIM Song et al. (2020) sampling, we also achieve a 3× faster inference, making our method practical for clinical use. Experiments on LIDC-IDRI (Armato III et al., 2011) show that TDG-DiffDPO sets a new state of the art in segmentation quality while preserving diversity, and introduces a controllable preference knob that allows practitioners to directly adjust the balance between per-sample accuracy and distributional variability.

## 1 Introduction

Medical image segmentation plays a central role in modern healthcare. It helps doctors in detecting disease, plan treatments, guide surgeries, and track patient progress. However, unlike natural images, medical images often exhibit inherent ambiguity: anatomical boundaries may be blurred, modalities like MRI and ultrasound suffer from noise and low contrast, and even experienced clinicians frequently disagree on the correct delineation of structures. As a result, multiple segmentation masks for a single image can be considered clinically valid. Most deep learning models, however, ignore this reality. They produce a single "best guess" mask, which may look accurate by metrics like Dice score but cannot capture the full range of clinically valid possibilities. This can give a false sense of certainty, especially in cases where uncertainty is actually part of the truth. Diffusion models offer a way forward. Because their generation process is inherently stochastic, they can produce not just one mask but a variety of plausible ones. In principle, this allows us to model the entire distribution of expert interpretations. Yet, the standard diffusion training objective maximizes data likelihood without explicitly encouraging clinical quality. As a result, they may generate diverse masks, but many of them fall short in terms of clinical validity. This challenge parallels alignment problems in large language models (LLMs), where raw likelihood training does not guarantee outputs consistent with human preferences. Recent work solved this through preference-based alignment methods like Direct Preference Optimization (DPO) (Rafailov et al., 2023), and even more recently, this idea has been extended to diffusion models (Wallace et al., 2023).

Inspired by this, we propose **Text- and Dice-Guided Diffusion-DPO (TDG-DiffDPO)**, a framework designed specifically for ambiguous medical segmentation. Instead of relying on costly human preference labels, we create preferences automatically by comparing how close different masks are

Figure 1: Text annotation corresponding to the above image is: `The annotation id is 128. The subtlety is 5. The internalStructure is 1. The calcification is 6. The sphericity is 5. The margin is 4. The lobulation is 1. The spiculation is 5. The texture is 4. The malignancy is 4...` Although there are descriptions of the concerned region, the requirement is to identify the segmentation boundaries. More detail in Appendix.

to multi-expert ground truths, using Dice similarity as the guiding signal. By fine-tuning the diffusion model with these preference pairs, we encourage it to favor masks that better reflect expert consensus while still preserving the natural diversity of clinically valid interpretations.

**Our key contributions include:**

- **First unified text-conditional diffusion framework for ambiguous medical segmentation:** We design a novel diffusion-based architecture that incorporates textual descriptions as conditioning inputs (see Figure 1), enabling the model to generate clinically guided and **ambiguity-aware** segmentations.

- **First DPO-based controlled alignment and novel training strategy for ambiguous segmentation:** We integrate domain-informed text prompts, simulated ambiguous regions, and diverse annotations so the model learns to produce a range of valid masks. To the best of our knowledge, we are the first to adapt preference optimization (DPO) to ambiguous medical segmentation, exploring consensus-based, expert-specific, and stochastic preference strategies.

- **Comprehensive experiments with state-of-the-art results and efficient deployment:** On the LIDC-IDRI dataset, our method achieves state-of-the-art results preserving distributional diversity while improving average Dice outperforming deterministic and stochastic baselines. By incorporating DDIM sampling Song et al. (2020), we achieve a $3\times$ speedup, making the approach suitable for large-scale and clinical use.

## 2 RELATED WORK

### 2.1 MEDICAL IMAGE SEGMENTATION

Medical image segmentation is central to computer-aided diagnosis, treatment planning, surgical guidance, and patient monitoring. Early approaches relied on classical image processing methods such as thresholding, edge detection, and region growing, but their robustness and generalization were limited by noise, low contrast, and anatomical variability. The rise of deep learning reshaped medical image segmentation. U-Net (Ronneberger et al., 2015) introduced an encoder-decoder architecture with skip connections, preserving spatial details crucial for accurate segmentation. Variants such as Attention U-Net, Residual U-Net, and Gated U-Net further improved performance by enhancing feature propagation and contextual awareness. More recently, transformers, which excel at modeling long-range dependencies, have been integrated into segmentation pipelines. TransUNet (Chen et al., 2021) incorporates transformers into the U-Net bottleneck, while UNETR (Hatamizadeh et al., 2022) moves them to the encoder stage, enabling direct 3D volumetric processing with global context.

## 2.2 AMBIGUITY IN MEDICAL IMAGE SEGMENTATION

One of the biggest challenges in medical image segmentation is ambiguity. Images are often noisy, blurred, or low in contrast, and anatomical structures can overlap in ways that make their boundaries hard to define. Even clinicians may disagree on the "correct" segmentation, as shown by multi-annotator datasets like LIDC-IDRI (Armato III et al., 2011). To capture this uncertainty, researchers have turned to probabilistic and generative models. Approaches such as the Probabilistic U-Net (Kohl et al., 2018) use conditional variational autoencoders to produce multiple plausible segmentations, while hierarchical models like PHi-Seg (Baumgartner et al., 2019) structure uncertainty at different levels, making outputs more interpretable. Other methods including ensembles, Bayesian deep learning with Monte Carlo dropout, and conditional GANs have also been explored. These approaches marked real progress, but often generated samples with limited diversity or clinical alignment. More recently, diffusion models have emerged as a compelling alternative. By iteratively denoising data to learn complex distributions, diffusion frameworks like CIMD (Rahman et al., 2023) capture the range of expert annotations and generate segmentation hypotheses that carry clinical meaning.

## 2.3 TEXT-GUIDED GENERATION FOR MEDICAL IMAGING

Text-guided generative models have shown how powerful language can be in shaping image generation. Systems like DALL·E, Stable Diffusion (Rombach et al., 2022), and Imagen reveal that natural language prompts are enough to produce realistic, high-quality images with remarkable detail and control. This idea is especially compelling in medical imaging, where domain knowledge already exists in the form of radiology reports and clinical notes. If segmentation models could tap into this knowledge through text, they wouldn't just generate masks blindly. Instead, they could be guided toward variations that clinicians actually consider when interpreting an image. By conditioning segmentation on text, models can capture clinically meaningful variations rather than producing ambiguous outputs without context. When combined with diffusion, text guidance offers a flexible way to control uncertain regions and ensures outputs remain grounded in clinical reasoning.

## 2.4 MODEL ALIGNMENT WITH PREFERENCE LEARNING

Generating multiple possible segmentations is useful, but it doesn't automatically make a model clinically valuable. For adoption in practice, the outputs also need to meet expert-defined quality standards. In NLP, this kind of "alignment" has often been achieved using reinforcement learning from human feedback (RLHF), but RLHF can be unstable and costly. A more recent alternative, Direct Preference Optimization (DPO) (Rafailov et al., 2023), reframes the problem as a simpler classification task over pairs of preferred outputs. Diffusion-DPO extended this into vision (Wallace et al., 2023), showing that diffusion models can be guided by human preferences through a tractable surrogate objective. In our work, we create preference data that encodes both clinical quality standards and how ambiguity should be handled. This allows the model to favor segmentations that are not only plausible, but also clinically meaningful. To the best of our knowledge, this is the first approach to unify probabilistic diffusion, text guidance, and preference alignment for segmentation.

## 3 METHODOLOGY

Our proposed framework, Text- and Dice-Guided Diffusion-DPO (TDG-DiffDPO), is structured as a two-stage process designed to first learn a rich, diverse distribution of plausible segmentations and then meticulously align this distribution with clinically relevant quality standards. The first stage involves training a text-conditional diffusion model for ambiguous segmentation, which we term **AmbiSeg**, to serve as our high-quality reference model, $p_{\text{ref}}$. The second, and core, stage involves fine-tuning AmbiSeg using our novel Dice-guided DPO objective to produce the final, aligned model, $p_\theta$. The overall architecture is depicted in Figures 2 and 3.

### 3.1 PRELIMINARIES: DENOISING DIFFUSION PROBABILISTIC MODELS

Our work is built upon the foundation of denoising diffusion probabilistic models (DDPMs) (Ho et al., 2020), a class of generative models that learn to reverse a fixed noise-adding process.

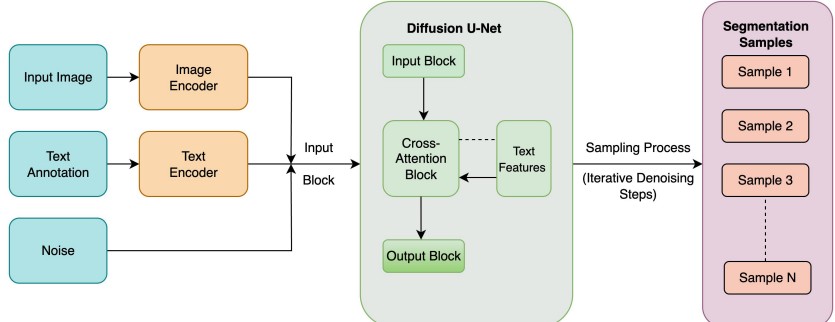

Figure 2: The AmbiSeg Architecture: A conditional diffusion framework that takes a medical image and a text prompt as input. The image provides spatial conditioning, while the text prompt, encoded by Bio_ClinicalBERT, guides the U-Net denoiser via cross-attention to generate ambiguity-aware segmentations.

### 3.1.1 FORWARD PROCESS (DIFFUSION)

The forward process, $q$, systematically corrupts an initial clean data sample $y_0$ (in our case, a segmentation mask) with Gaussian noise over $T$ discrete timesteps. This process is defined as a Markov chain where the noise at each step is governed by a predefined variance schedule $\{\beta_t\}_{t=1}^T$:

$$q(y_t|y_{t-1}) = \mathcal{N}(y_t; \sqrt{1-\beta_t}y_{t-1}, \beta_t\mathbf{I}). \tag{1}$$

A significant property of this process is that we can sample a noisy version $y_t$ directly from the original clean sample $y_0$ in a single step, using a closed-form expression:

$$q(y_t|y_0) = \mathcal{N}(y_t; \alpha_t y_0, \sigma_t^2\mathbf{I}), \tag{2}$$

where $\alpha_t = \sqrt{\bar{\alpha}_t}$, $\sigma_t = \sqrt{1-\bar{\alpha}_t}$, and $\bar{\alpha}_t = \prod_{i=1}^t(1-\beta_i)$. This allows for efficient training by sampling arbitrary timesteps $t$ without iterating through the entire chain.

### 3.1.2 REVERSE PROCESS (DENOISING)

The generative power of a diffusion model lies in its ability to learn the reverse process, $p_\theta(y_{t-1}|y_t)$. This process starts with pure Gaussian noise, $y_T \sim \mathcal{N}(0, \mathbf{I})$, and iteratively denoises it to produce a clean sample, $y_0$. The model, parameterized by $\theta$, is typically a neural network $\epsilon_\theta(y_t, t)$ that is trained to predict the noise component $\epsilon$ that was added to create $y_t$. The training objective is a simplified variational bound that reduces to a mean squared error loss between the true and predicted noise:

$$\mathcal{L}_{\text{DDPM}} = \mathbb{E}_{y_0, \epsilon \sim \mathcal{N}(0,\mathbf{I}), t}\left[||\epsilon - \epsilon_\theta(\alpha_t y_0 + \sigma_t\epsilon, t)||^2\right]. \tag{3}$$

## 3.2 STAGE 1: AMBISEG, THE TEXT-CONDITIONAL REFERENCE MODEL

The effectiveness of DPO alignment is contingent on a capable reference model. Therefore, we first develop **AmbiSeg**, a powerful text-conditional diffusion model designed specifically for ambiguous segmentation.

### 3.2.1 ARCHITECTURAL DETAILS

The core of AmbiSeg, depicted in Figure 2, is a conditional U-Net denoiser, $\epsilon_{\text{ref}}(y_t, t, b, c)$, which takes the noisy mask $y_t$, the timestep $t$, the input medical image $b$, and a text prompt $c$ as input.

**Input Representation and Conditioning.** Our model's diffusion process applies **only to the segmentation mask**, not the image itself. At each forward timestep $t$, Gaussian noise is added only to the mask channel $\boldsymbol{y}_0$ to produce a noisy mask $\boldsymbol{y}_t$. The input medical image $b$ is normalized and treated as a fixed spatial condition throughout the denoising process. The U-Net denoiser receives the input image and the noisy mask concatenated along the channel dimension, forming the input

tensor $z_t = [b, y_t]$. This architecture ensures that the model learns the distribution of valid segmentations conditioned on a static visual context.

**Text Conditioning via Cross-Attention.** The core of our model is a U-Net denoiser with a standard encoder-decoder structure, enhanced with residual blocks and self-attention modules. To enable fine-grained semantic control, we incorporate textual guidance using a cross-attention mechanism. First, the text prompt $c$ (e.g., "A large, spiculated nodule with solid texture") is encoded into a 768-dimensional embedding using a pre-trained, frozen **Bio_ClinicalBERT** model (Alsentzer et al., 2019). These text embeddings are then integrated into the decoder blocks of the U-Net. At each resolution level, the spatial feature maps serve as the **Query (Q)** vectors, while the text embedding is linearly projected to form the **Key (K)** and **Value (V)** vectors. The attention operation is then performed as:

$$\text{Attention}(Q, K, V) = \text{softmax}\left(\frac{QK^T}{\sqrt{d_k}}\right) V, \tag{4}$$

where $d_k$ is the dimension of the key vectors. This mechanism allows the model to dynamically attend to different parts of the text prompt to modulate the spatial features, effectively "steering" the generation process.

### 3.2.2 TRAINING PROTOCOL FOR AMBIGUITY

To explicitly model ambiguity from multi-rater annotations, we train AmbiSeg with a hybrid loss function. At each training step, we uniformly sample one expert mask $y_0$ for a given image. The loss combines a standard denoising objective with a KL-divergence term, which encourages the model to learn a distribution over possible segmentations. The final objective is:

$$\mathcal{L}_{\text{AmbiSeg}} = \mathcal{L}_{\text{mse}} + \lambda \mathcal{L}_{\text{kl}}. \tag{5}$$

The MSE term is the standard noise prediction loss: $\mathcal{L}_{\text{mse}} = \mathbb{E}[||\epsilon - \epsilon_{\text{ref}}(\alpha_t y_0 + \sigma_t \epsilon, t, b, c)||^2]$. The KL term, $\mathcal{L}_{\text{kl}} = \text{KL}(q_\phi(y|b, c)||p_\psi(y|b, c))$, regularizes the model's predictions to match a learned prior and posterior distribution over segmentations conditioned on the image. This dual objective forces the model to learn the full multi-modal distribution of plausible annotations. The model parameters are updated using gradient descent on this total loss until convergence.

### 3.3 INFERENCE FOR DIVERSE SEGMENTATION GENERATION

During inference, AmbiSeg leverages its generative design to produce diverse segmentation masks, each representing a different but plausible interpretation of the input image $b$. We generate 16 proposals by combining variation in text prompts with randomness in sampling. An annotation CSV, created by extracting metadata and descriptive information from the dataset, provides diverse text prompts $c$ that guide the model toward different semantic perspectives. For each prompt, the reverse diffusion process is initialized with a random noise sample $y_T \sim \mathcal{N}(\mathbf{0}, \mathbf{I})$, ensuring that even identical prompts can yield distinct outputs. Together, text guidance and stochastic initialization capture the ambiguity inherent in medical imaging, producing a spectrum of clinically plausible segmentations. The generation unfolds over $T$ reverse diffusion steps, gradually refining noisy inputs into coherent masks that reflect both the visual content and the semantic cues from text.

### 3.4 STAGE 2: TDG-DIFFDPO ALIGNMENT

With a capable reference model, we proceed to the core of our contribution: aligning AmbiSeg's generative distribution towards higher clinical quality using a novel Dice-guided DPO framework.

### 3.4.1 REVISITING THE DIFFUSION-DPO OBJECTIVE

As the standard DPO objective is intractable for diffusion models, we adopt the Diffusion-DPO formulation. The loss function, adapted for our conditional segmentation task, is:

$$\mathcal{L}_{\text{TDG-DiffDPO}}(\theta) = -\mathbb{E}_{(b,c,y^+,y^-)\sim\mathcal{D},t,\epsilon}\left[\log \sigma(-\beta \cdot \Delta_\theta(y_t^+, y_t^-, b, c))\right], \tag{6}$$

where $\sigma$ is the sigmoid function, $\beta$ is a temperature parameter, and the crucial $\Delta_\theta$ term measures the relative improvement of the policy model $\epsilon_\theta$ over the reference model $\epsilon_{\text{ref}}$ on the winning $(y^+)$ and

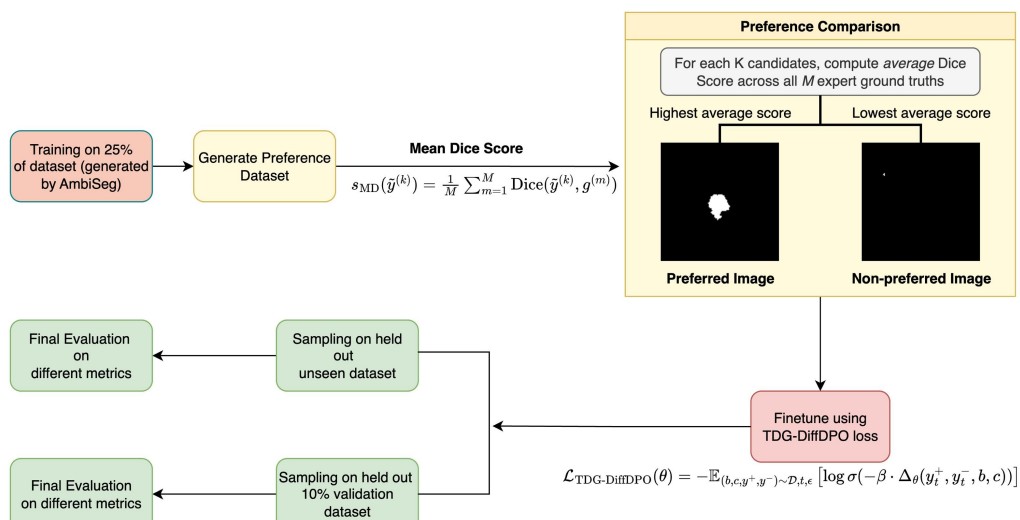

Figure 3: The TDG-DiffDPO Framework: Stage 1 trains the reference model (AmbiSeg). In Stage 2, we generate candidate masks, construct preference pairs $(y^+, y^-)$ using a Dice-based strategy (e.g., Mean Dice), and fine-tune the model with the DPO objective to align its outputs with clinical consensus.

losing ($y^-$) samples:

$$\Delta_\theta(y_t^+, y_t^-, b, c) = \underbrace{\left(||\epsilon - \epsilon_\theta(y_t^+, t, b, c)||^2 - ||\epsilon - \epsilon_{\text{ref}}(y_t^+, t, b, c)||^2\right)}_{\text{Improvement on winner } y^+}$$
$$- \underbrace{\left(||\epsilon - \epsilon_\theta(y_t^-, t, b, c)||^2 - ||\epsilon - \epsilon_{\text{ref}}(y_t^-, t, b, c)||^2\right)}_{\text{Improvement on loser } y^-}. \qquad (7)$$

This objective encourages the model to reduce the denoising error on winning samples more significantly than on losing samples.

### 3.4.2 NOVELTY: DICE-GUIDED PREFERENCE CONSTRUCTION STRATEGIES

The primary bottleneck for DPO is the acquisition of preference data. Our innovation is to fully automate this process using a clinically relevant metric. We designed and ablated three distinct strategies, each testing a specific hypothesis about what constitutes an optimal preference signal. For each strategy, we first sample $K$ candidates from the reference model and then apply a scoring rule to select the winner $y^+$ and loser $y^-$.

**Best-Rater Match (The Greedy Hypothesis).** This method hypothesizes that the best sample is one that achieves a near-perfect match with *at least one* expert, regardless of what other experts think. It directly aims to teach the model to "try to perfectly imitate one of the experts," thereby optimizing for the $D_{\max}$ metric. For each of the $K$ candidate masks, we find its single best Dice score by comparing it against all $M$ expert ground truths. The candidate with the highest of these "best-rater" scores is the winner; the one with the lowest is the loser. The scoring function is $s_{\text{BR}}(\tilde{y}^{(k)}) = \max_{m \in \{1,..,M\}} \text{Dice}(\tilde{y}^{(k)}, g^{(m)})$.

**Mean Dice (The Consensus Hypothesis).** This method values consensus. It hypothesizes that the best sample is not an extreme but a good *compromise or average* of all expert opinions. It aims to teach the model to "generate a mask that is the most agreeable and representative of the entire group of experts." For each of the $K$ candidates, we calculate its *average* Dice score across all $M$ expert ground truths. The candidate with the highest average score is the winner; the one with the lowest is the loser. The scoring function is $s_{\text{MD}}(\tilde{y}^{(k)}) = \frac{1}{M} \sum_{m=1}^{M} \text{Dice}(\tilde{y}^{(k)}, g^{(m)})$.

**Stochastic Run Preference (SRP) (The Exploration Hypothesis).** This method explores the model's own potential. It hypothesizes that the strongest preference signal comes from contrasting the absolute best possible ("lucky") outcome and the absolute worst possible ("unlucky") outcome the model is capable of producing. It teaches the model to "be more like your best possible self and less like your worst possible self." We perform two independent sampling runs (with different random seeds), creating a larger pool of $2K$ masks. From this combined pool, we find the single mask with the absolute highest Best-Rater score to be the winner, and the one with the absolute lowest score to be the loser. This uses the $s_{\text{BR}}$ scoring rule but on an expanded candidate set generated from multiple stochastic runs to identify the model's performance extremes. After constructing the preference dataset $\mathcal{D}$ using one of these strategies, we initialize the policy model $\epsilon_\theta$ with the weights of the frozen reference model $\epsilon_{\text{ref}}$ and fine-tune it using the TDG-DiffDPO loss.

## 3.5 Efficient Inference with DDIM

To address the significant computational cost of standard DDPM sampling (often requiring $T = 1000$ steps), we employ Denoising Diffusion Implicit Models (DDIM) Song et al. (2020) for inference. DDIM formulates a non-Markovian generative process that allows for a deterministic mapping from a latent variable to a sample. This enables us to use a much shorter sampling trajectory (e.g., 100 steps) with a minimal loss in sample quality, providing a crucial 3x speed-up that makes our framework practical for extensive evaluation and future clinical use.

## 4 Experiments

We conduct a series of rigorous experiments to validate the effectiveness of our proposed TDG-DiffDPO framework. Our evaluation is designed to answer several key research questions: 1) Can DPO-based alignment, guided by a clinical metric, improve the quality of a stochastic segmentation model? 2) Which preference construction strategy provides the most effective alignment signal? 3) What is the trade-off between per-sample quality and distributional diversity after alignment? 4) Can our method achieve state-of-the-art performance compared to existing approaches for ambiguous segmentation?

### 4.1 Datasets and Preprocessing

We conduct our experiments on the widely recognized **LIDC-IDRI** (Armato III et al., 2011) dataset, a benchmark for thoracic CT scans with detailed lung nodule annotations from four board-certified radiologists. Unlike some previous works that focus on selected 2D lesion slices, our approach includes *all* nodules, regardless of size. This broader inclusion is made possible by our text-guidance mechanism. We curate custom text prompts from radiology reports and scan metadata, explicitly describing attributes like nodule size and texture. These descriptions guide the diffusion model during training and inference, enabling it to better handle the full spectrum of nodule types, including small or ambiguous ones. By not relying on manual slice selection or size-based filtering, we train on a more comprehensive dataset. We evaluate on all 3,072 nodules in the test set, generating 16 diverse segmentation samples per image. The full dataset and code are provided in the supplementary material. A detailed description of our text prompt curation process can be found in Appendix A.1.

### 4.2 Evaluation Metrics

To evaluate our models, we use a comprehensive suite of metrics detailed in Appendix A.3.

**Ambiguity-Aware Metrics.** To assess the quality of the generated distribution, we use **Generalized Energy Distance (GED)**, which measures the similarity between predicted and ground-truth distributions, and the **Collective Insight (CI) Score**, a harmonic mean of Combined Sensitivity ($S_c$), Maximum Dice Score ($D_{\max}$), and Diversity Agreement (DA). Lower GED and higher CI are better.

**Standard Metrics.** To measure the success of DPO alignment on per-sample quality, we report standard overlap metrics, including the **Average Dice Score** and **Average Intersection over Union (IoU)** over the generated samples.

### 4.3 IMPLEMENTATION DETAILS

Our AmbiSeg model uses a U-Net denoiser with self-attention and cross-attention modules. The reference model was trained for 50,000 steps using AdamW with a linear learning rate schedule. For DPO alignment, we fine-tuned the reference model on preference datasets constructed from the training data. For all final evaluations, we generated $N = 16$ samples per image using a DDIM sampler Song et al. (2020) with 100 inference steps for efficiency. This choice of $N = 16$ is justified by an ablation study in Appendix A.2, which shows it provides an optimal balance between diversity and quality. Complete architectural specifications, training hyperparameters, and the DPO evaluation protocol are detailed in Appendix A.1.

## 5 RESULTS

We present both quantitative and qualitative evaluations of our method. All experiments are conducted on the LIDC-IDRI dataset, unless otherwise stated.

Table 1: Baseline performance and DDIM acceleration (without DPO). DDIM achieves a $3\times$ speedup with a minor trade-off in performance, making it practical for large-scale experiments.

| Method | GED ↓ | CI ↑ | $D_{max}$ ↑ | Avg. inference time (N=16) |
|---|---|---|---|---|
| AmbiSeg (DDPM, 1000 steps) | 0.152 | 0.836 | 0.814 | ∼69h (4 GPUs) |
| AmbiSeg (DDIM, 100 steps) | 0.177 | 0.809 | 0.783 | ∼23h (4 GPUs) |

### 5.1 COMPARISON WITH BASELINES

Table 2 compares our method with probabilistic and diffusion-based baselines. Our AmbiSeg reference model, even before DPO alignment, establishes a strong baseline, significantly outperforming prior methods like CIMD (Rahman et al., 2023) and PHi-Seg (Baumgartner et al., 2019) on ambiguity-aware metrics. This highlights the effectiveness of our text-conditional architecture. The qualitative results in Figure 4 further illustrate that our model generates more diverse and clinically plausible segmentations compared to baselines.

Table 2: Quantitative comparison on LIDC-IDRI. Our AmbiSeg model serves as a strong baseline, outperforming prior work. TDG-DiffDPO further improves quality (see Table 3).

| Method | GED ↓ | CI ↑ | $D_{max}$ ↑ |
|---|---|---|---|
| Probabilistic U-Net (Kohl et al., 2018) | 0.353 | 0.731 | 0.892 |
| PHI-Seg (Baumgartner et al., 2019) | 0.270 | 0.736 | 0.904 |
| Generalized Prob. U-Net (Bhat et al., 2022) | 0.299 | 0.707 | 0.905 |
| CIMD (Rahman et al., 2023) | 0.321 | 0.759 | 0.915 |
| AmbiSeg (DDIM, Ours) | 0.177 | 0.809 | 0.783 |
| **TDG-DiffDPO (Ours, Final)** | **0.206** | **0.746** | **0.785** |

### 5.2 FINAL MODEL PERFORMANCE AND DPO ABLATION

To determine the best preference strategy, we conducted an ablation study detailed in Appendix A.4.2. The **Mean Dice** strategy, which rewards consensus, proved most effective. We selected this for our final TDG-DiffDPO model.

Table 3 shows the final performance on the test set. TDG-DiffDPO achieves a new state-of-the-art **Average Dice score of 0.398**, significantly outperforming the strong AmbiSeg baseline (0.318). This demonstrates that our DPO alignment successfully improves the clinical quality of generated samples. As expected, there is a trade-off: the GED increases slightly from 0.177 to 0.206. This reflects

Figure 4: Qualitative comparison with baselines on LIDC-IDRI. Each example shows the input image and four expert annotations (ground truth). For fair comparison, we display the first four segmentation samples from each model. Note that empty masks are valid annotations, reflecting expert disagreement. Our method aims to capture this variability, as seen in the third row where some experts provided no segmentation.

that the model's distribution has been intentionally narrowed to favor higher-quality, consensus-aligned masks, which is the desired outcome of preference alignment. Figure 7 provides a visual comparison of samples before and after DPO alignment.

Table 3: Final model comparison on the full test set ($N = 16$). Our TDG-DiffDPO model, aligned with the Mean Dice strategy, sets a new state of the art in per-sample quality (Avg. Dice/IoU) while maintaining strong distributional properties.

| Method | Avg. Dice ↑ | Avg. IoU ↑ | GED ↓ | CI ↑ | $D_{max}$ ↑ | Sc ↑ | DA ↑ |
|---|---|---|---|---|---|---|---|
| CIMD | 0.298 | 0.277 | 0.306 | 0.470 | 0.684 | 0.582 | 0.835 |
| DDIM Baseline | 0.318 | 0.266 | **0.177** | **0.809** | 0.783 | 0.748 | **0.866** |
| Mean Dice (25%) | **0.398** | **0.336** | 0.206 | 0.746 | **0.785** | **0.756** | 0.857 |
| Mean Dice (50%) | 0.380 | 0.345 | 0.237 | 0.720 | 0.768 | 0.742 | 0.859 |
| Stochastic Run | 0.387 | 0.348 | 0.210 | 0.749 | 0.783 | 0.761 | 0.862 |

## 5.3 Inference Efficiency

As shown in Table 1, our use of DDIM sampling Song et al. (2020) provides a significant practical advantage. By reducing the number of inference steps from 1000 to 100, we achieve a $3\times$ speedup in generation time with only a marginal trade-off in metric performance. This efficiency makes our framework feasible for large-scale analysis and brings it closer to clinical applicability.

Some additional results on relationship between mean DICE and GED score, and effect of DPO is presented in Appendix.

## 6 Conclusion

We have presented TDG-DiffDPO, a novel framework that successfully aligns a diffusion model for ambiguous medical image segmentation with clinically relevant quality metrics. Our core innovation: the use of a scalable, Dice-guided preference signal for DPO proved highly effective. Through rigorous ablation, we identified that a consensus-based preference signal is key to robustly improving segmentation quality. Our final model achieves state-of-the-art performance in average sample quality on the LIDC-IDRI benchmark, establishing a new paradigm for building controllable, high-quality generative models for medicine. By integrating DDIM for practical inference and demonstrating a principled trade-off between diversity and accuracy, TDG-DiffDPO represents a significant step towards creating more reliable and clinically aligned tools for medical image analysis.

## Ethics Statement

We have followed the ICLR Code of Ethics throughout this work. Our study makes use of the LIDC-IDRI dataset, a publicly available and anonymized collection of medical images, and therefore did not require additional Institutional Review Board (IRB) approval. The objective of our research is

to enhance the reliability of medical image segmentation tools, which could have a positive societal impact by supporting clinical diagnosis. We do not anticipate any direct negative societal consequences from this work. Nonetheless, as with any machine learning system designed for healthcare applications, extensive clinical validation would be essential before real-world deployment.

REPRODUCIBILITY STATEMENT

To promote reproducibility, we provide detailed documentation of our methodology and make all necessary resources available. The full source code for our TDG-DiffDPO framework including scripts for preprocessing, model training, DPO fine-tuning, and evaluation is included in the supplementary material. All experiments were conducted on the publicly available LIDC-IDRI dataset, with patient-level data splits and preprocessing procedures described in Appendix A.1. Comprehensive details of the model architecture, training hyperparameters, and DPO evaluation protocol are also reported in Appendix A.1. In addition, both the ambiguity-aware and standard evaluation metrics are formally defined in Appendix A.3.

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

## A APPENDIX

### A.1 IMPLEMENTATION AND EXPERIMENTAL DETAILS

**Dataset Splits and Preprocessing.** We follow a subject-level split to prevent data leakage between training and test sets. From the 1012 subjects in the LIDC-IDRI dataset, 80% were used for training and 20% for testing. All grayscale CT slices were resized to $128 \times 128$ pixels and normalized to the range $[-1, 1]$. Slices with no annotations from any of the four radiologists were treated as having empty masks.

**Text Prompt Curation.** To provide meaningful semantic guidance, we curated descriptive text prompts by systematically extracting and structuring metadata from the LIDC-IDRI dataset. The raw metadata for each nodule includes several radiologist-provided ratings (e.g., subtlety, sphericity, texture) on a 1-5 scale. Our curation process involved converting these ratings into natural language sentences using a predefined template. For instance, a 'spiculation' rating of 5 was translated to "The Spiculation is Marked Spiculation." These sentences were concatenated to form a comprehensive descriptive paragraph for each nodule. This structured approach ensures that the textual guidance is consistent, grounded in clinical data, and rich in semantic detail. An example prompt is provided in Figure 1.

**Diffusion Model and Training.** We used a diffusion process with $T = 1000$ timesteps and a linear noise schedule for $\beta_t$ where $\beta_1 = 10^{-4}$ and $\beta_T = 0.02$. The denoising backbone is a U-Net with a base channel size of 128, with multi-head self-attention and cross-attention modules (64 channels per head). The reference model was trained for 50,000 steps with a batch size of 32 using the AdamW optimizer and a linear learning rate decay from 1e-4. We used a composite loss $\mathcal{L}_{\text{total}} = \mathcal{L}_{\text{MSE}} + \lambda \mathcal{L}_{\text{KL}}$, with $\lambda = 0.001$.

**Text Encoder.** We used the Bio_ClinicalBERT (Alsentzer et al., 2019) model (Alsentzer et al., 2019) as a frozen text encoder. Prompts were tokenized and passed through BERT to obtain 768-dimensional embeddings, which were then linearly projected to match the 128-dimensional channel size in the U-Net's attention layers.

**TDG-DiffDPO Fine-tuning and Evaluation Protocol.** For DPO alignment, we first trained a baseline AmbiSeg model on 25% of the training data. Using this checkpoint as our reference model, we generated preference datasets for each of our three strategies (Best-Rater Match, Mean Dice, SRP) by sampling $K = 4$ candidate masks per image from the same 25% training split. We fine-tuned the checkpoint on each preference dataset using a constant learning rate of 5e-7 for 3 epochs with $\beta_{DPO} = 5000$. We empirically found 3 epochs to be the most effective training schedule

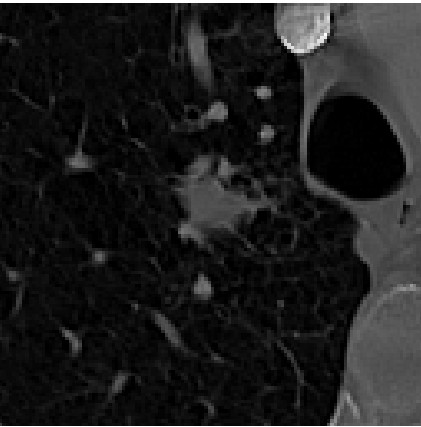

Figure 5: Text annotation corresponding to the above image is: `The annotation id is 128. The subtlety is 5. The internalStructure is 1. The calcification is 6. The sphericity is 5. The margin is 4. The lobulation is 1. The spiculation is 5. The texture is 4. The malignancy is 4. The Subtlety is Obvious. The InternalStructure is Soft Tissue. The Calcification is Absent. The Sphericity is Round. The Margin is Near Sharp. The Lobulation is No Lobulation. The Spiculation is Marked Spiculation. The Texture is Solid/Mixed. The Malignancy is Moderately Suspicious. The diameter mm is 31.92. The volume mm`$^3$` is 11568.45. The slice indices is [101, 102, 103, 104, 105, 106, 107, 108, 109, 110, 111, 112, 113]. The bbox is (slice(np.int64(264), np.int64(315), None), slice(np.int64(173), np.int64(217), None), slice(np.int64(101), np.int64(114), None)). Average malignancy (meta): 5.00. Is cancer (any slice): True. Is clean (all slices): False. Diameter: 20.6846910857402 mm. Malignancy level: 5.0. Malignancy (nodule): 1.0.` Although there are descriptions of the concerned region, the requirement is to identify the segmentation boundaries.

for DPO fine-tuning. Longer schedules led to overtraining, which introduced noise and degraded alignment quality. The best strategy (Mean Dice) was identified on a 10% validation set. The final model reported in the main paper is this very same checkpoint, evaluated on the completely unseen test set by generating $N = 16$ samples per image. An Exponential Moving Average (EMA) of model weights with a decay of 0.9999 was used in all training stages. As an additional experiment, we trained on 50% of the training data and then fine-tuned using our best DPO strategy, Mean-Dice, to evaluate whether performance improves with a larger portion of training data.

**Inference.** For all final results, we generated $N = 16$ segmentation samples per image using a DDIM sampler Song et al. (2020) with 100 inference steps. Model outputs, initially in the range $[-1, 1]$, were binarized at a threshold of 0.5 for evaluation.

A.2 ANALYSIS OF THE NUMBER OF GENERATED SAMPLES

An important aspect of our experimental setup is the number of samples ($N$) generated per input image. To justify our choice and investigate the trade-off between diversity and computational cost, we performed an ablation study by varying $N$ from 1 to 32. The goal was to identify when additional samples stop contributing meaningfully to capturing the ground-truth ambiguity.

The results, shown in Table 4, reveal a clear trend. For lower values of $N$ (1 to 8), performance improves significantly, highlighting that a single deterministic output is insufficient for modeling diagnostic uncertainty. As $N$ increases to 16, most key metrics **CI Score**, **Combined Sensitivity**, $D_{\mathbf{max}}$, and **Diversity Agreement** either peak or approach saturation, indicating comprehensive coverage of the expert annotation space.

Beyond $N = 16$, however, we observe a slight decline in performance. Metrics such as the **CI Score** and $D_{\mathbf{max}}$ decrease, while **GED** increases, indicating a weaker alignment with the target distribution. A closer qualitative inspection reveals that many of the additional samples beyond this point are blank, offering no meaningful contribution. These degenerate outputs effectively introduce noise into the distribution. This analysis confirms that $\mathbf{N = 16}$ strikes an optimal balance. As such, $N = 16$ is used throughout all main experiments, consistent with findings from prior work (Rahman et al., 2023).

Table 4: Ablation on the number of generated samples ($N$). Performance generally peaks or saturates around $N = 16$, justifying its use for robust evaluation.

| num_samples | CI $\uparrow$ | Sc $\uparrow$ | $D_{\mathbf{max}} \uparrow$ | DA $\uparrow$ | GED $\downarrow$ |
|---|---|---|---|---|---|
| 1 | 0.246 | 0.347 | 0.388 | 0.575 | 0.796 |
| 2 | 0.374 | 0.520 | 0.519 | 0.527 | 0.190 |
| 4 | 0.586 | 0.672 | 0.641 | 0.768 | 0.179 |
| 8 | 0.736 | 0.809 | 0.735 | 0.851 | 0.180 |
| 12 | 0.776 | 0.848 | 0.768 | 0.861 | 0.183 |
| **16** | **0.801** | **0.876** | **0.790** | **0.866** | **0.179** |
| 24 | 0.795 | 0.871 | 0.782 | 0.858 | 0.184 |
| 32 | 0.790 | 0.868 | 0.775 | 0.855 | 0.188 |

### A.3 DETAILED EVALUATION METRICS

To provide a comprehensive and rigorous evaluation of our model, we employ a suite of metrics designed to capture different facets of performance in an ambiguous segmentation setting. These metrics are divided into two main categories: (1) **Standard Metrics**, which assess the geometric accuracy of a single prediction against a single ground truth, and (2) **Ambiguity-Aware Metrics**, which evaluate the quality of the entire distribution of generated segmentations.

#### A.3.1 STANDARD METRICS

These metrics are foundational in segmentation tasks but are inherently limited in our context because they do not account for valid inter-observer variability. A model could produce a "perfect" average mask that receives a high score, yet fail to capture the diverse, clinically plausible segmentations provided by different experts. We include them primarily for completeness and comparison with deterministic methods.

**Dice Coefficient (Dice)** The Dice Coefficient is one of the most common metrics in medical image segmentation. It measures the overlap between two binary masks, $X$ (prediction) and $Y$ (ground truth), and can be interpreted as twice the area of overlap divided by the total number of pixels in both masks. It is particularly sensitive to the size of the segmented region. A score of 1 indicates perfect overlap, while 0 indicates no overlap.

$$\text{Dice}(X, Y) = \frac{2|X \cap Y|}{|X| + |Y|}. \tag{8}$$

**Intersection over Union (IoU)** Also known as the Jaccard Index, IoU measures the extent of overlap between the predicted mask $X$ and the ground truth mask $Y$. It is defined as the ratio of the area of their intersection to the area of their union. Like Dice, it ranges from 0 to 1, with 1 signifying a perfect match.

$$\text{IoU}(X, Y) = \frac{|X \cap Y|}{|X \cup Y|}. \tag{9}$$

#### A.3.2 AMBIGUITY-AWARE METRICS

These metrics are central to our evaluation as they are designed to assess a model's ability to generate a distribution of outputs that is both accurate and diverse, reflecting the inherent ambiguity in the ground truth. Let $\mathcal{S} = \{s_1, s_2, \ldots, s_n\}$ be the set of $n$ segmentation masks generated by our model

for a given input, and let $\mathcal{G} = \{g_1, g_2, \ldots, g_m\}$ be the set of $m$ ground truth masks provided by multiple expert annotators.

**Combined Sensitivity ($S_c$)** This metric answers the question: "Does the model's set of predictions collectively cover all the areas that **any** expert annotated?" It measures the recall of the union of all generated masks with respect to the union of all ground truth masks. A high $S_c$ indicates that the model's generated distribution does not miss any regions considered relevant by the pool of experts.

$$S_U = \bigcup_{i=1}^{n} s_i \quad \text{and} \quad G_U = \bigcup_{j=1}^{m} g_j. \tag{10}$$

$$S_c = \frac{|S_U \cap G_U|}{|G_U|}. \tag{11}$$

**Maximum Dice Score ($D_{\mathbf{max}}$)** This metric addresses the question: "For each expert's annotation, does the model produce at least one prediction that is a good match?" For every ground truth mask in $\mathcal{G}$, it finds the single best-matching sample from the generated set $\mathcal{S}$ (based on Dice score) and then averages these best-match scores. A high $D_{\max}$ demonstrates that the model is capable of generating specific predictions that align well with each individual expert's opinion, rather than just producing a generic average.

$$D_{\max} = \frac{1}{|\mathcal{G}|} \sum_{g \in \mathcal{G}} \max_{s \in \mathcal{S}} \text{Dice}(s, g). \tag{12}$$

**Diversity Agreement (DA)** This metric seeks to answer: "Does the amount of disagreement among the model's predictions match the amount of disagreement among the experts?" It quantifies how well the diversity within the generated samples aligns with the diversity observed in the ground truths. It does this by comparing the range of dissimilarities (e.g., $1 - \text{Dice}$) in the generated set to the range in the ground truth set. It penalizes models that are either under-diverse (mode collapse) or over-diverse (generating unrealistic variations). A DA score close to 1 indicates a perfect match in the level of uncertainty.

$$\text{DA} = 1 - \frac{|\Delta V_{\min}| + |\Delta V_{\max}|}{2}, \tag{13}$$

where $V_{\min}$ and $V_{\max}$ represent the minimum and maximum pairwise dissimilarities within the ground truth (GT) and sample ($S$) sets, respectively, and $\Delta V$ is their difference.

### A.3.3 COMPOSITE AMBIGUITY METRICS

These metrics combine the lower-level ambiguity measures into single, powerful scores that provide a holistic view of distributional quality.

**Generalized Energy Distance (GED)** GED provides a formal measure of the distance between two distributions. In our case, it compares the distribution of predicted segmentations ($P_X$) with the distribution of ground truth segmentations ($P_Y$). The formula balances three terms: (1) the average distance between predictions and ground truths, (2) the average distance among predictions (a measure of generated diversity), and (3) the average distance among ground truths (a measure of expert diversity). A low GED score indicates that the model's output distribution is a close match to the expert distribution in terms of both location and spread.

$$\text{GED}^2(P_X, P_Y) = 2\,\mathbb{E}[d(X, Y)] - \mathbb{E}[d(X, X')] - \mathbb{E}[d(Y, Y')], \tag{14}$$

where $X, X' \sim P_X$, $Y, Y' \sim P_Y$, and $d(\cdot, \cdot)$ is a distance metric ($1 - \text{Dice}$).

**Collective Insight (CI) Score** To provide a single, interpretable score, the CI Score computes the harmonic mean of the three key ambiguity metrics: Combined Sensitivity ($S_c$), Maximum Dice Score ($D_{\max}$), and Diversity Agreement (DA). The use of a harmonic mean is critical because it heavily penalizes poor performance in any single component. A model cannot achieve a high CI score by excelling at coverage ($S_c$) while failing at diversity (DA); it must perform well across all three dimensions of coverage, accuracy, and diversity simultaneously.

$$\text{CI} = \frac{3 \times S_c \times D_{\max} \times \text{DA}}{S_c \cdot D_{\max} + D_{\max} \cdot \text{DA} + S_c \cdot \text{DA}}. \tag{15}$$

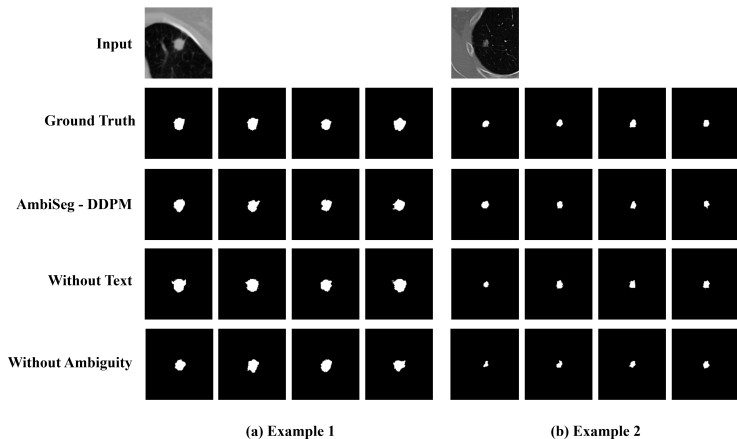

(a) Example 1        (b) Example 2

Figure 6: **Ablation analysis on the LIDC-IDRI dataset comparing *AmbiSeg* against its variants.** The full AmbiSeg model better reflects expert-level variability, while removing text or the ambiguity objective leads to overconfident or less diverse outputs.

## A.4 ABLATION STUDIES

### A.4.1 ABLATION ON AMBISEG COMPONENTS

To validate the design of our reference model, AmbiSeg, we performed ablation studies (Table 5). Removing text guidance ('w/o Text') leads to a noticeable drop in performance, particularly in GED, indicating the model struggles to generate segmentations that align well with the ground-truth distribution. Removing the explicit ambiguity modeling objective ('w/o Ambiguity', i.e., the $\mathcal{L}_{kl}$ term) results in mode collapse, reflected in a sharp decline in the CI score and Diversity Agreement. These results, also visualized in Figure 6, confirm that both text guidance and the hybrid loss are critical for modeling diagnostic uncertainty.

Table 5: Ablation study of AmbiSeg components on the full test set. Both text guidance and the ambiguity modeling objective are crucial for performance.

| Method | CI $\uparrow$ | Sc $\uparrow$ | $D_{max}$ $\uparrow$ | DA $\uparrow$ | GED $\downarrow$ |
|---|---|---|---|---|---|
| AmbiSeg (Full Model - DDPM) | **0.835** | **0.895** | **0.814** | **0.885** | **0.152** |
| AmbiSeg (Full Model - DDIM) | **0.808** | **0.748** | **0.782** | **0.866** | **0.177** |
| w/o Text | 0.766 | 0.876 | 0.743 | 0.864 | 0.214 |
| w/o Ambiguity | 0.716 | 0.773 | 0.731 | 0.851 | 0.173 |

### A.4.2 ABLATION ON PREFERENCE CONSTRUCTION FOR DPO

We evaluated our three preference construction strategies on a validation set to select the best alignment signal (Table 6). The **Mean Dice** strategy delivered the most significant improvements in per-sample quality, boosting the Average Dice from 0.308 to 0.407. While Best-Rater Match and SRP also showed gains, the consensus-based signal from Mean Dice proved most effective at guiding the model towards segmentations that are broadly agreeable to all experts, which was our primary goal for alignment.

## A.5 CHOICE OF PIXEL-SPACE DIFFUSION.

Our model performs the diffusion process directly in the pixel space rather than a compressed latent space, as seen in Latent Diffusion Models (LDMs) (Rombach et al., 2022). While LDMs offer significant computational efficiency, we opted for a pixel-space model for two primary reasons. First, medical image segmentation requires extremely high fidelity and preservation of fine-grained details, such as subtle texture and irregular boundaries, which can sometimes be lost or distorted during the

Table 6: DPO alignment: ablation on preference construction strategies ($N = 4$). The baseline DDIM was trained on 25% of data, fine-tuned with each strategy, and evaluated on a 10% validation set. Mean Dice shows the best improvement in Dice/IoU.

| Method | Avg. Dice ↑ | Avg. IoU ↑ | GED ↓ | CI ↑ | $D_{max}$ ↑ | Sc ↑ | DA ↑ |
|---|---|---|---|---|---|---|---|
| DDIM Baseline | 0.308 | 0.255 | **0.178** | 0.559 | 0.605 | **0.686** | 0.715 |
| Best-Rater Match | 0.301 | 0.246 | 0.193 | **0.565** | 0.592 | 0.718 | 0.715 |
| Mean Dice | **0.407** | **0.366** | 0.215 | 0.514 | **0.670** | 0.506 | **0.745** |
| Stochastic Run | 0.389 | 0.340 | 0.223 | 0.558 | 0.677 | 0.544 | 0.749 |

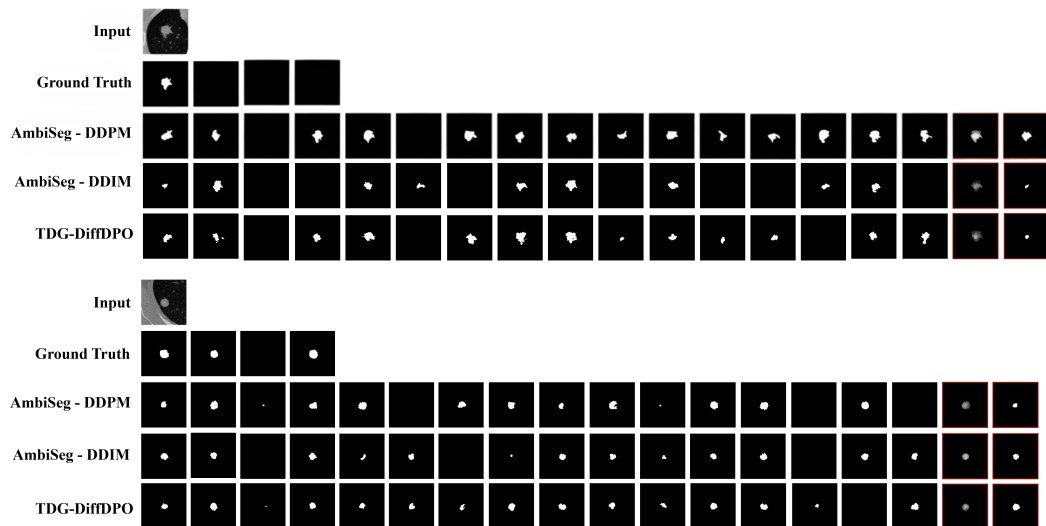

Figure 7: Example of diverse segmentation samples generated by AmbiSeg for a lung nodule. The figure shows the input CT scan, ground truth annotations from four experts, 16 unique samples from our model, the ensemble average, and the final majority vote segmentation. The effect of DPO alignment is also visualized.

compression and decompression stages of an LDM's autoencoder. Second, by operating directly on the pixels, our model avoids introducing an additional source of architectural complexity and potential information loss, allowing for a more direct study of the effects of text guidance and ambiguity modeling. We acknowledge the computational cost as a limitation and view the integration of these principles into an efficient LDM framework as a promising direction for future work.

## A.6 ADDITIONAL QUALITATIVE RESULTS

We provide additional qualitative examples to showcase our model's capabilities. Figure 7 shows the 16 diverse samples generated by our model for a given input, illustrating its ability to capture inter-observer variability. Figure 8 provides another head-to-head comparison with CIMD, further highlighting the superior diversity and quality of our AmbiSeg and TDG-DiffDPO models. Finally,

## A.7 QUALITATIVE VALIDATION OF SEMANTIC CONTROL

While our quantitative results demonstrate the model's overall performance, a crucial aspect of our work is its ability to interpret and act upon the semantic content of text prompts. To better validate this capability, we carried out a focused qualitative experiment to clearly test the impact of textual guidance. We chose two challenging test cases with notable expert disagreement in their ground truth annotations and conditioned our model on contradictory descriptive prompts for the same input image. Figure 9 demonstrates the fine-grained semantic control offered by our text-conditioning mechanism.

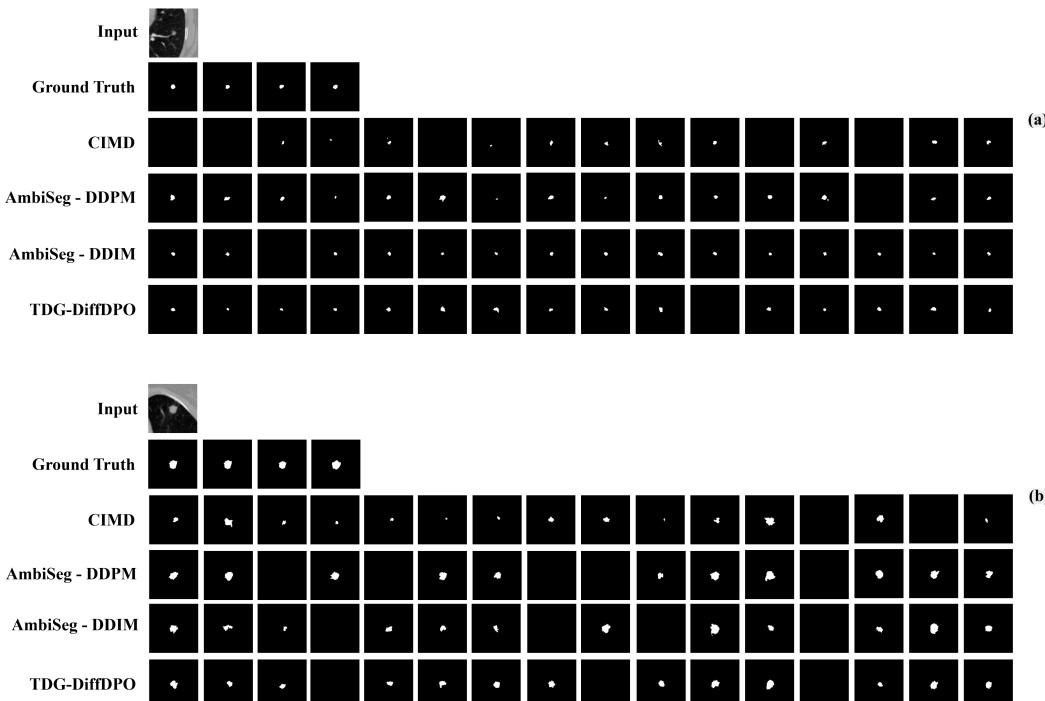

Figure 8: Additional qualitative comparison on lung nodule segmentation. Each input CT slice is paired with four expert-provided ground truth annotations. AmbiSeg and TDG-DiffDPO generate a diverse set of plausible masks that better reflect the expert variability compared to the baseline.

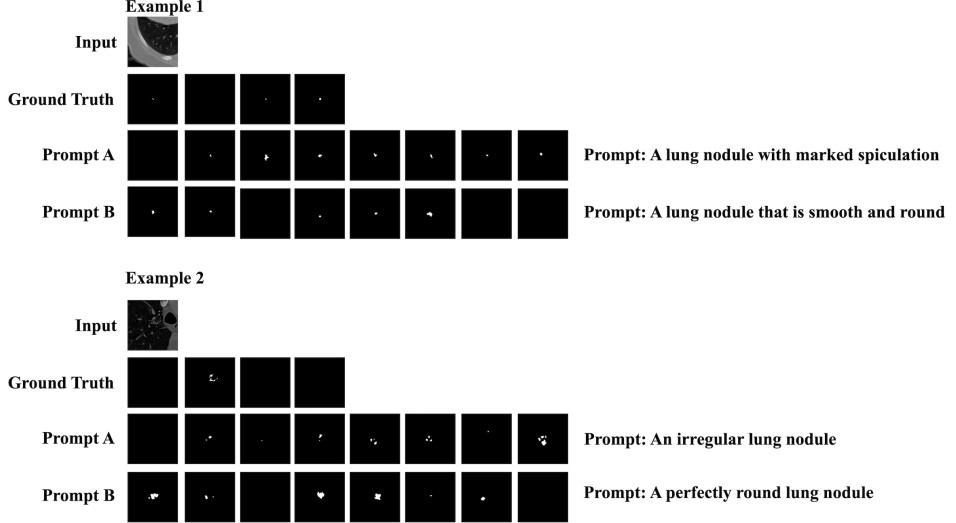

Figure 9: Qualitative validation of semantic control across two distinct examples. For each input image (top row of each example), we generate 8 segmentation samples using both a factually descriptive prompt (Prompt A) and a semantically contradictory prompt (Prompt B), with expert-annotated ground truths provided for reference. The model's outputs consistently adapt to the textual description, producing shapes with the requested geometric properties (e.g., spiculated vs. smooth in Example 1; irregular vs. round in Example 2), demonstrating its ability to interpret and act on semantic guidance while respecting underlying image ambiguity.

**Example 1: Control over Nodule Texture ('Spiculated' vs. 'Smooth')**

The first case, the Figure 9 features a highly subtle nodule where expert annotations varied greatly, with one radiologist providing an empty mask. The textual metadata from the dataset for this nodule is particularly revealing; for instance, the full prompt from one of the annotation includes the descriptions: 'The Subtlety is Fairly Subtle. The Sphericity is Ovoid/Round. The Spiculation is No Spiculation.'. Such descriptions explicitly capture the uncertainty in how the nodule should be interpreted.

We first conditioned the model with a prompt aligned to one possible interpretation: "A lung nodule with marked spiculation." (Prompt A). The generated samples reflect the inherent ambiguity, spanning from an entirely empty mask (matching the dissenting expert) to several small segmentations with jagged, irregular contours consistent with spiculation.

Next, using the same input image, we provided a semantically contradictory prompt: "A lung nodule that is smooth and round." (Prompt B). The model responded clearly, producing segmentations with more compact shapes and smoother, rounded boundaries; directly adapting its output to the new textual guidance.

**Example 2: Control over Nodule Margin ('Irregular' vs. 'Round')**

To test the consistency of this effect, the second case, the Figure 9 presents a visually complex and irregular region where three of four experts provided empty masks. Here, the textual metadata strongly aligns with the visual evidence. The full prompt for one of the annotation includes key descriptors such as: 'The Margin is Near Poorly Defined. The Lobulation is Marked Lobulation. The Spiculation is Marked Spiculation.'.

When guided by a factual prompt summarizing this description: "An irregular lung nodule" (Prompt A): the model's outputs reflected expert consensus. Most samples were empty, and the few non-empty masks were highly fragmented and irregular, consistent with the "irregular" characterization.

The stronger test came with the contradictory prompt: "A perfectly round lung nodule" (Prompt B). Despite the visually complex evidence, the model adapted to the textual instruction, generating masks that were noticeably simpler and more rounded than those from Prompt A. This demonstrates the model's ability to enforce semantic constraints from text even when they conflict with the underlying image data.

Overall, these experiments demonstrate strong evidence of genuine semantic control. The model not only interprets high-level concepts (e.g., "spiculated," "round") but also translates them into the corresponding geometric characteristics of its segmentations. By grounding the evaluation in the dataset's full descriptive prompts, we show that the model can generate outputs aligned with factual descriptions and, more importantly, flexibly adjust when given contradictory semantic instructions. This highlights the central contribution of our work: a controllable and ambiguity-aware segmentation model.

## THEORETICAL RELATIONSHIP BETWEEN GED AND DICE

We recall the Generalized Energy Distance (GED) between the distribution of predicted segmentations $P_X$ and the distribution of ground-truth segmentations $P_Y$:

$$\text{GED}^2(P_X, P_Y) = 2\,\mathbb{E}[d(X,Y)] - \mathbb{E}[d(X,X')] - \mathbb{E}[d(Y,Y')], \tag{16}$$

where $X, X' \sim P_X$ are independent predictions, $Y, Y' \sim P_Y$ are independent ground-truth masks, and $d(\cdot, \cdot)$ is a distance metric. In our work we adopt

$$d(y,g) = 1 - \text{Dice}(y,g).$$

*Proposition* 1 (Dice decomposition of GED). Let $P_X$ denote the distribution of predicted segmentations and $P_Y$ the distribution of ground-truth segmentations. Suppose the base distance in the GED is chosen as

$$d(y,g) \;=\; 1 - \text{Dice}(y,g), \qquad \text{Dice}(y,g) \in [0,1].$$

Then the squared Generalized Energy Distance admits the equivalent form

$$\text{GED}^2(P_X, P_Y) \;=\; \mathbb{E}_{X,X'\sim P_X}\big[\text{Dice}(X,X')\big] \tag{17}$$

$$+ \mathbb{E}_{Y,Y'\sim P_Y}\big[\text{Dice}(Y,Y')\big] \tag{18}$$

$$- 2\,\mathbb{E}_{X\sim P_X,\,Y\sim P_Y}\big[\text{Dice}(X,Y)\big]. \tag{19}$$

*Proof.* Start from the definition (Eq. equation 16):

$$\text{GED}^2(P_X, P_Y) = 2\,\mathbb{E}_{X \sim P_X, Y \sim P_Y}[d(X, Y)] - \mathbb{E}_{X, X' \sim P_X}[d(X, X')] - \mathbb{E}_{Y, Y' \sim P_Y}[d(Y, Y')].$$

Substitute $d(\cdot, \cdot) = 1 - \text{Dice}(\cdot, \cdot)$. Using linearity of expectation we get

$$\text{GED}^2(P_X, P_Y) = 2\,\mathbb{E}_{X,Y}[1 - \text{Dice}(X, Y)] - \mathbb{E}_{X,X'}[1 - \text{Dice}(X, X')]$$
$$- \mathbb{E}_{Y,Y'}[1 - \text{Dice}(Y, Y')]$$

$$= \big(2\,\mathbb{E}_{X,Y}[1] - 2\,\mathbb{E}_{X,Y}[\text{Dice}(X, Y)]\big)$$
$$- \big(\mathbb{E}_{X,X'}[1] - \mathbb{E}_{X,X'}[\text{Dice}(X, X')]\big)$$
$$- \big(\mathbb{E}_{Y,Y'}[1] - \mathbb{E}_{Y,Y'}[\text{Dice}(Y, Y')]\big).$$

Since expectations of the constant 1 equal 1, and there are two such constants in the first term and one each in the second and third, collect constants and Dice terms:

$$\text{GED}^2(P_X, P_Y) = (2 \cdot 1 - 1 - 1)$$
$$- \big(2\,\mathbb{E}_{X,Y}[\text{Dice}(X, Y)] - \mathbb{E}_{X,X'}[\text{Dice}(X, X')] - \mathbb{E}_{Y,Y'}[\text{Dice}(Y, Y')]\big)$$

$$= 0$$
$$- \big(2\,\mathbb{E}_{X,Y}[\text{Dice}(X, Y)] - \mathbb{E}_{X,X'}[\text{Dice}(X, X')] - \mathbb{E}_{Y,Y'}[\text{Dice}(Y, Y')]\big).$$

Therefore,

$$\text{GED}^2(P_X, P_Y) = \mathbb{E}_{X,X'}[\text{Dice}(X, X')] + \mathbb{E}_{Y,Y'}[\text{Dice}(Y, Y')] - 2\,\mathbb{E}_{X,Y}[\text{Dice}(X, Y)].$$

$\square$

*Proposition* 2 (Deterministic predictions). If the model collapses to a single segmentation $x^*$, i.e. $P_X = \delta_{x^*}$, then

$$\text{GED}^2(\delta_{x^*}, P_Y) = 1 + \mathbb{E}_{Y,Y' \sim P_Y}\big[\text{Dice}(Y, Y')\big] - 2\,\mathbb{E}_{Y \sim P_Y}\big[\text{Dice}(x^*, Y)\big]. \tag{3}$$

Consequently, for fixed $P_Y$, minimizing $\text{GED}^2(\delta_{x^*}, P_Y)$ over $x^*$ is equivalent to maximizing the average Dice overlap $\mathbb{E}_{Y \sim P_Y}[\text{Dice}(x^*, Y)]$.

*Proof.* Start from the Dice-form GED decomposition (Proposition 1):

$$\text{GED}^2(P_X, P_Y) = \mathbb{E}_{X,X' \sim P_X}\big[\text{Dice}(X, X')\big] + \mathbb{E}_{Y,Y' \sim P_Y}\big[\text{Dice}(Y, Y')\big] - 2\,\mathbb{E}_{X \sim P_X, Y \sim P_Y}\big[\text{Dice}(X, Y)\big].$$

If $P_X = \delta_{x^*}$, then every sample $X \sim P_X$ equals $x^*$ almost surely. Hence

$$\mathbb{E}_{X,X' \sim P_X}\big[\text{Dice}(X, X')\big] = \text{Dice}(x^*, x^*) = 1,$$

because the Dice of a mask with itself is 1. Also

$$\mathbb{E}_{X \sim P_X, Y \sim P_Y}\big[\text{Dice}(X, Y)\big] = \mathbb{E}_{Y \sim P_Y}\big[\text{Dice}(x^*, Y)\big].$$

Substituting these into the decomposition yields

$$\text{GED}^2(\delta_{x^*}, P_Y) = 1 + \mathbb{E}_{Y,Y' \sim P_Y}\big[\text{Dice}(Y, Y')\big] - 2\,\mathbb{E}_{Y \sim P_Y}\big[\text{Dice}(x^*, Y)\big],$$

which is exactly (3). $\square$

*Remark* (**Equivalence of objectives**). For fixed $P_Y$ the term $\mathbb{E}_{Y,Y' \sim P_Y}[\text{Dice}(Y, Y')]$ is constant with respect to $x^*$. Therefore, minimizing $\text{GED}^2(\delta_{x^*}, P_Y)$ over $x^*$ is equivalent to maximizing $\mathbb{E}_{Y \sim P_Y}[\text{Dice}(x^*, Y)]$, i.e., the average Dice overlap between the single predicted mask and the annotator distribution.

*Proposition* 3 (Tradeoff between Dice and GED). From Eq. equation 19, GED depends on two competing terms:

- the *cross-term* $\mathbb{E}_{X,Y}[\text{Dice}(X, Y)]$, which increases with per-sample segmentation quality;

- the *within-term* $\mathbb{E}_{X,X'}[\text{Dice}(X, X')]$, which increases as model predictions become less diverse.

Formally,

$$\text{GED}^2(P_X, P_Y) = E_{X,X'} + E_{Y,Y'} - 2E_{X,Y},$$

where $E_{X,Y} = \mathbb{E}_{X,Y}[\text{Dice}(X, Y)]$. Increasing per-sample Dice ($E_{X,Y}$) reduces GED, while reducing prediction diversity (raising $E_{X,X'}$) increases GED unless accompanied by a sufficiently large gain in $E_{X,Y}$.

**Corollary 1** (Consensus vs. distributional alignment). *Let $b \in \mathbb{R}^{C \times H \times W}$ denote an input image with expert masks $G(b) = \{g^{(m)}\}_{m=1}^{M}$. For a candidate prediction $\tilde{y}^{(k)} \sim p_\theta(\cdot|b)$, the* Mean Dice *strategy scores*

$$s_{\text{MD}}(\tilde{y}^{(k)}) = \frac{1}{M}\sum_{m=1}^{M} \text{Dice}(\tilde{y}^{(k)}, g^{(m)}). \tag{20}$$

*This consensus hypothesis maximizes per-sample overlap but ignores prediction diversity, and therefore may yield high Dice yet poor GED. In contrast, GED explicitly rewards both high per-sample Dice and distributional alignment with annotator variability.*

In summary, Dice evaluates single-sample overlap, while GED generalizes Dice to the distributional setting. Optimizing Dice alone encourages mode collapse, whereas minimizing GED enforces both per-sample quality and correct distributional spread.

## DIFFUSION-DPO WITH MEAN DICE PREFERENCES

We construct preferences by scoring each candidate $\tilde{y}^{(k)} \sim p_\theta(\cdot|b)$ using its average Dice overlap with all annotators:

$$s_{\text{MD}}(\tilde{y}^{(k)}) = \frac{1}{M}\sum_{m=1}^{M} \text{Dice}(\tilde{y}^{(k)}, g^{(m)}).$$

For each pair $(y^+, y^-)$ with $s_{\text{MD}}(y^+) > s_{\text{MD}}(y^-)$, we form a preference tuple. The Diffusion-DPO loss is unchanged:

$$\mathcal{L}_{\text{DPO}}(\theta) = -\log\sigma\left(\beta\log\frac{p_\theta(y^+|b)}{p_{\text{ref}}(y^+|b)} - \beta\log\frac{p_\theta(y^-|b)}{p_{\text{ref}}(y^-|b)}\right).$$

**Theorem 1** (DPO upweights consensus-aligned samples). *If $s_{\text{MD}}(y^+) > s_{\text{MD}}(y^-)$, minimizing $\mathcal{L}_{DPO}$ enforces*

$$\frac{p_\theta(y^+|b)}{p_\theta(y^-|b)} > \frac{p_{ref}(y^+|b)}{p_{ref}(y^-|b)}.$$

*Thus, Diffusion-DPO shifts likelihood toward predictions that achieve higher* average Dice *across all annotators, i.e. consensus-agreeable masks.*

*Proposition* 4 (Impact on GED under Mean Dice ranking). Recall

$$\text{GED}^2(P_X, P_Y) = E_{X,X'} + E_{Y,Y'} - 2E_{X,Y},$$

where $E_{X,Y} = \mathbb{E}_{X \sim P_X, Y \sim P_Y}[\text{Dice}(X, Y)]$. Since preference construction is based on mean Dice, DPO explicitly increases the expected cross-term $E_{X,Y}$ measured against the empirical annotator distribution $P_Y$. As before, diversity effects enter through $E_{X,X'}$: if DPO collapses predictions to a single consensus mask, $E_{X,X'} \to 1$ and GED increases; if diversity is preserved while improving mean Dice, GED decreases.

**Remark:**(Upper bound under consensus preferences) From nonnegativity of GED,

$$E_{X,Y} \leq \tfrac{1}{2}\left(E_{X,X'} + E_{Y,Y'}\right).$$

Since Diffusion-DPO with mean Dice preferences increases $E_{X,Y}$, the achievable per-sample average Dice is bounded by the annotator self-consistency $E_{Y,Y'}$. In practice, tasks with low inter-annotator agreement limit the Dice gains achievable via preference optimization.

In summary, Diffusion-DPO trained with *Mean Dice* preferences upweights consensus-agreeable predictions. This improves average Dice scores by construction, but its effect on GED depends on whether the model preserves distributional diversity or collapses to a single consensus prediction.

## PREFERENCE STRATEGIES AND THEIR EFFECT ON DICE / GED

**Notation**   For an image $b$ let $G(b) = \{g^{(m)}\}_{m=1}^M$ be annotator masks and $\{\tilde{y}^{(k)}\}_{k=1}^K \sim p_\theta(\cdot \mid b)$ be an i.i.d. candidate set drawn from the model. Denote by $P_X$ the model predictive distribution (after training) and by $P_Y$ the annotator distribution. Define

$$E_{X,Y} = \mathbb{E}_{X \sim P_X, Y \sim P_Y}[\mathrm{Dice}(X,Y)], \qquad E_{X,X'} = \mathbb{E}_{X,X' \sim P_X}[\mathrm{Dice}(X,X')].$$

*Proposition* 5 (Mean Dice maximizes consensus cross-Dice among candidates). Fix an image $b$ and a finite candidate pool $\{\tilde{y}^{(k)}\}_{k=1}^K$. Define the Mean-Dice score

$$s_{\mathrm{MD}}(\tilde{y}^{(k)}) = \frac{1}{M} \sum_{m=1}^M \mathrm{Dice}(\tilde{y}^{(k)}, g^{(m)}).$$

Let $\tilde{y}^\star = \arg\max_k s_{\mathrm{MD}}(\tilde{y}^{(k)})$ be the Mean-Dice winner and let $x^\star$ denote a single-prediction policy that always returns $\tilde{y}^\star$ (i.e. $P_X = \delta_{\tilde{y}^\star}$). Then among all single-candidate deterministic policies that choose one of the $K$ candidates, $\tilde{y}^\star$ maximizes the expected Dice to the annotator distribution restricted to $G(b)$:

$$\tilde{y}^\star = \arg\max_k \ \mathbb{E}_{Y \sim \hat{P}_Y}[\mathrm{Dice}(\tilde{y}^{(k)}, Y)],$$

where $\hat{P}_Y$ is the empirical annotator distribution that places mass $1/M$ on each $g^{(m)}$. Consequently, for the deterministic GED formula

$$\mathrm{GED}^2(\delta_{\tilde{y}}, \hat{P}_Y) = 1 + E_{Y,Y'} - 2\mathbb{E}_{Y \sim \hat{P}_Y}[\mathrm{Dice}(\tilde{y}, Y)],$$

the Mean-Dice winner $\tilde{y}^\star$ is the minimizer of $\mathrm{GED}^2$ over the $K$ deterministic candidates.

*Proof.* By definition,

$$\mathbb{E}_{Y \sim \hat{P}_Y}[\mathrm{Dice}(\tilde{y}^{(k)}, Y)] = \frac{1}{M} \sum_{m=1}^M \mathrm{Dice}(\tilde{y}^{(k)}, g^{(m)}) = s_{\mathrm{MD}}(\tilde{y}^{(k)}).$$

Therefore the argument that maximizes the mean-Dice score also maximizes the empirical expected Dice against $\hat{P}_Y$, proving the first claim.

For any deterministic candidate $\tilde{y}$ we have (Prop. 2)

$$\mathrm{GED}^2(\delta_{\tilde{y}}, \hat{P}_Y) = 1 + E_{Y,Y'} - 2\mathbb{E}_{Y \sim \hat{P}_Y}[\mathrm{Dice}(\tilde{y}, Y)].$$

Since $E_{Y,Y'}$ and the constant 1 are independent of $\tilde{y}$, the GED is minimized exactly by maximizing $\mathbb{E}_{Y \sim \hat{P}_Y}[\mathrm{Dice}(\tilde{y}, Y)]$, which, as shown, is achieved by $\tilde{y}^\star$. $\qquad\square$

