# OpenReview forum: "Controllable Preference Alignment for Ambiguous Medical Image Segmentation via Text and Dice Guidance"
_ICLR.cc/2026/Conference — Submitted to ICLR 2026_

### Official Review · Reviewer_5GeQ · 2025-10-30

**Soundness:** 3
**Presentation:** 3
**Contribution:** 3
**Rating:** 4
**Confidence:** 4

**Summary:**

This paper transfers preference alignment (DPO) to the setting of ambiguous medical image segmentation, uses multi-annotator Dice to automatically construct preference pairs, and couples it with a text-conditioned reference diffusion model (AmbiSeg).

**Strengths:**

(1) This paper presents a diffusion-based architecture that incorporates textual descriptions as conditioning inputs, enabling the model to generate clinically guided and ambiguity-aware segmentations.

(2) This paper adapts preference optimization (DPO) to ambiguous medical segmentation, exploring consensus-based, expert-specific, and stochastic preference strategies.

(3) Extensive experimental results show the effectiveness of the propsoed framework.

**Weaknesses:**

(1) Quantitative comparisons do not explicitly mark the best/second-best results (e.g., bold/underline/shaded cells), reducing table informativeness and comparability.

(2) The manuscript repeatedly emphasizes a controllable preference knob, but the Methods section does not define this knob’s specific variable, semantics, or scope (which parameter family it belongs to, its tunable range, and how it affects the objective).

(3) In the two-stage framework, Stage-2 (TDG-DiffDPO) directly adopts an existing Diffusion-DPO objective; no new training form is introduced. Overall, it reads as “porting DPO to segmentation with textual conditioning and constructing preferences via Dice scores,” rather than proposing a new preference-optimization principle or diffusion-alignment framework.

(4) All experiments are conducted on the single LIDC-IDRI dataset; the lack of cross-center/cross-modality or external validation makes it difficult to substantiate the claimed stability and generality of “controllable alignment” under distribution shift.

**Questions:**

(1) Figure 1 is not cited in the main text, which harms narrative completeness and retrievability.

(2) Notation is inconsistent—for example, in Sec.3.2, the image is first denoted by b and later by x; the notation should be unified throughout to avoid ambiguity.

(3) Effects of the parameter $\lambda$ should be investigated.

---

> ### Author Response · Authors · 2025-11-15
>
> We thank the reviewer for their feedback. Here are our replies to the weaknesses and questions posed.
>
> 1. Quantitative comparisons do not explicitly mark the best/second-best results (e.g., bold/underline/shaded cells), reducing table informativeness and comparability.
>
> Reply: Thanks for pointing it out. We will mark the best/second-best results in our updated manuscript.
>
> 2. The manuscript repeatedly emphasizes a controllable preference knob, but the Methods section does not define this knob’s specific variable, semantics, or scope (which parameter family it belongs to, its tunable range, and how it affects the objective).
>
> Reply: The preference knob or the choice corresponds to the various metrics used to create the preference data. In our case, the preference strategies are Mean Dice, Stochastic Run and Best Rater Match (Refer Table 6 in supplementary for comparisons).  These strategies are based on Dice Score. The control also depends on the amount of training data used for DPO finetuning, as well as the number of epochs it is trained for. In Table 3, in main paper, we show the effect of amount of training data on the resulting preference optimization and their corresponding effect on metrics such as Dice and GED.
>
> 3. In the two-stage framework, Stage-2 (TDG-DiffDPO) directly adopts an existing Diffusion-DPO objective; no new training form is introduced. Overall, it reads as “porting DPO to segmentation with textual conditioning and constructing preferences via Dice scores,” rather than proposing a new preference-optimization principle or diffusion-alignment framework.
>
> Reply: Our novelty is not in proposing a new DPO objective but rather proposing a novel metric based preference optimization strategy for multimodal "ambiguous" segmentation. In Table 6, Page 16, we show the comparison of metrics for various preference strategies we proposed. Our work highlights the importance of metric based preference optimization unlike other alignment strategies where human labels or scores such as CLIP/Aesthetics (Classical Image generation as in DiffusionDPO) are sometimes used. Our empirical observations demonstrate that our novel proposed preference optimization strategies are suitable for alignment of "ambiguous" medical image segmentation with respect to Dice Score, GED scores and corresponding text.
>
> 4. All experiments are conducted on the single LIDC-IDRI dataset; the lack of cross-center/cross-modality or external validation makes it difficult to substantiate the claimed stability and generality of “controllable alignment” under distribution shift.
>
> Reply: To the best of our knowledge, this is the "only" dataset which has both multi-annotator as well as metadata. We used this metadata to curate textual annotations. In supplementary Page 20, Theorem 1, we prove that the preference optimization approach necessarily shifts likelihood toward predictions that achieve higher average Dice across all annotators, i.e. consensus-agreeable mask. Unfortunately, empirically we are limited by single dataset. This theoretical result is independent of the dataset and hence expected to hold under distribution shift. Moreover, for cross-modality validation, we show in Figure 6 and Table 5 in supplementary. the effectiveness of text as a modality in our ambiguous image segmentation tasks. We found that with text, the scores improved which can be seen visually as well as in the table.
>
> 5. Figure 1 is not cited in the main text, which harms narrative completeness and retrievability.
>
> Reply: Thanks for pointing this out. We will make sure to cite the Figure 1 in our updated manuscript.
>
> 6. Notation is inconsistent—for example, in Sec.3.2, the image is first denoted by b and later by x; the notation should be unified throughout to avoid ambiguity.
>
> Reply: Thanks for pointing this out. We will make sure to correct the notation in our updated manuscript.
>
> 7. Effects of the parameter $\lambda$ should be investigated.
>
> Reply: Although ablation with $\lambda$ is not reported in the paper, our choice of $\lambda$ value was selected based on many experiments. Lower $\lambda$ value makes the KL loss term very small, steering the total loss towards MSE, thus not capturing enough ambiguity. Taking a large $\lambda$ value did result in good amount of ambiguity but at the cost of MSE. The value of  $\lambda$ we chose led us to the optimal Dice and GED scores. If required, we will be glad to add this reasoning in the paper.

---

### Official Review · Reviewer_iAA7 · 2025-10-31

**Soundness:** 3
**Presentation:** 4
**Contribution:** 3
**Rating:** 6
**Confidence:** 3

**Summary:**

This work proposes a two-stage framework for ambiguous medical image segmentation that generates a distribution of plausible masks using diffusion models. In the first stage, AmbiSeg is trained as a text-conditioned diffusion model that leverages image–text pairs to produce multiple diverse segmentation hypotheses reflecting inter-rater variability. In the second stage, TDG-DiffDPO fine-tunes AmbiSeg using Dice-guided Direct Preference Optimization, aligning the generated mask distribution toward clinically preferred segmentations based on multi-expert annotations. Experiments on the LIDC-IDRI lung CT dataset demonstrate that the proposed model achieves higher per-sample Dice and improved clinical alignment compared to prior probabilistic and diffusion-based baselines such as Probabilistic U-Net, PHi-Seg, and CIMD, while maintaining strong diversity and 3× faster inference with DDIM sampling.

**Strengths:**

1. The authors extend Diffusion-DPO to medical segmentation by using Dice similarity as the evaluation metric for preference alignment and explore three implementation strategies for constructing preference pairs. This adaptation enables automatic, clinically relevant supervision without human preference labels.
2. The idea of conditioning on text is promising, as shown both quantitatively (Table 5) and qualitatively (Section A.7).

**Weaknesses:**

1. Encoding short, structured metadata phrases with Bio-ClinicalBERT appears unnecessarily complex; simpler embeddings could achieve comparable results with lower cost. The inclusion of annotation IDs as conditioning inputs is not well justified and may risk introducing sample-specific bias rather than genuine semantic conditioning.
2. Despite improved ambiguity modeling and controllability, the model’s Dice scores remain lower than competing deterministic or diffusion-based baselines, suggesting a trade-off where preference alignment improves diversity but reduces segmentation precision.
3. All experiments are conducted solely on the LIDC-IDRI lung CT dataset.

**Questions:**

1. Could the authors evaluate the necessity of Bio-ClinicalBERT by testing a simpler text encoder or adding an ablation to show its impact on performance?
2. Can the authors include experiments on additional datasets to demonstrate generalizability, for example by following Probabilistic U-Net’s setup to simulate ambiguity on datasets like Cityscapes?
3. Could the authors provide a deeper analysis on why the Mean-Dice preference strategy performs best, and under what conditions the other strategies fail?
4. (Optional) It would be interesting to see a quantitative evaluation of text adherence—for instance, training a classifier on metadata and testing whether generated masks align with the corresponding text descriptions.

---

> ### Author Response · Authors · 2025-11-15
>
> We thank the reviewer for their feedback. Here are our replies to the weaknesses and questions posed.
>
> 1. Encoding short, structured metadata phrases with Bio-ClinicalBERT appears unnecessarily complex; simpler embeddings could achieve comparable results with lower cost. The inclusion of annotation IDs as conditioning inputs is not well justified and may risk introducing sample-specific bias rather than genuine semantic conditioning.
>
> Reply: The paper "Publicly Available Clinical BERT Embeddings" by Emily Alsentzer et al. introduced Bio-ClinicalBERT and they found out that BERT trained on clinical data has a higher MedNLI accuracy compared to BERT not trained on clinical data. If we use a simpler embedding architecture, we need to pretrain on clinical data. Not training it on clinical data, is expected to give a lower accuracy as demonstrated in the paper cited above. The effectiveness of smaller model and subsequent training is a topic suitable for future explorations. Bio-Clinical BERT creates prompts from the annotation data we curated, so it doesn't really take into account the annotation IDs as conditioning inputs, so there is no risk of sample specific bias. Moreover, the annotation ID's of the validation samples are not correlated to annotation ID's of training samples.
>
>
> 2. Despite improved ambiguity modeling and controllability, the model’s Dice scores remain lower than competing deterministic or diffusion-based baselines, suggesting a trade-off where preference alignment improves diversity but reduces segmentation precision.
>
> Reply: With our preference optimization approach, we demonstrate that one can increase the Dice score. In fact, we provide a controllable way of balancing Dice score with GED score. We observe that there is a tradeoff between Dice score and GED score. The GED score captures the closeness/similarity between the distribution of the generated segmentation and distribution of the 4 ground truth segmentations, whereas Dice score measures the overlap between predicted and ground-truth segmentation masks. Increasing Dice score often comes at a cost of worse GED score which is not necessarily a favorable outcome to capture "ambiguity".  Moreover, in Proposition 1 in supplementary, we theoretically prove the relationship/trade-off between Dice and GED score.
>
> 3. All experiments are conducted solely on the LIDC-IDRI lung CT dataset.
>
> Reply: To the best of our knowledge, this is the "only" dataset which has both multi-annotator as well as metadata. We used this metadata to curate textual annotations.
>
> 4. Could the authors evaluate the necessity of Bio-ClinicalBERT by testing a simpler text encoder or adding an ablation to show its impact on performance?
>
> Reply: Please refer to reply in Question 1.
>
> 5. Can the authors include experiments on additional datasets to demonstrate generalizability, for example by following Probabilistic U-Net’s setup to simulate ambiguity on datasets like Cityscapes?
>
> Reply: Cityscapes dataset does not have text annotations which is a crucial requirement to test our method. Most datasets either lack multi-annotator, or textual metadata.
>
> 6. Could the authors provide a deeper analysis on why the Mean-Dice preference strategy performs best, and under what conditions the other strategies fail?
>
> Reply: In supplementary, in pages 20 and 21, we theoretically show the effect of Preference strategies on Dice / GED. Moreover, in Supplementary in Table 6, we show the ablation of various preference optimization strategies namely Mean Dice, Stochastic Run, Best Rater Match and their corresponding performance metrics. We found out that Mean Dice is the best preference strategy and in the main paper, in Table 3, we show our final model comparison on the full test set with the Mean Dice strategy (trained on 25 % as well as 50 % of train data).

---

### Official Review · Reviewer_SCoE · 2025-10-31

**Soundness:** 2
**Presentation:** 2
**Contribution:** 2
**Rating:** 2
**Confidence:** 4

**Summary:**

The paper introduces TDG-DiffDPO to enable segmentation via text and dice guidance. Experiments are performed on LIDC-IDRI datasets.

**Strengths:**

1. The paper is well-motivated by the key challenges in text-guided medical image segmentation.
2. The paper presents both quantitative and qualitative results to demonstrate the outperformance of TDG-DiffDPO.

**Weaknesses:**

1. The architecture of TDG-DiffDPO looks very similar to previous work, for example, latent diffusion model, and DiffSeg.
2. The experiment only performs on one testbed (LIDC-IDRI). The generalizability remains a concern.
3. Several significant segmentation baselines are missing, for example, nnunet.
4. What is the segmentation performance of the other baseline?
5. Using the DDIM for faster sampling is hard to be considered as a novelty.

**Questions:**

N/A

---

> ### Author Response · Authors · 2025-11-15
>
> We thank the reviewer for their feedback. Here are our replies for the weaknesses pointed out:
>
> 1. The architecture of TDG-DiffDPO looks very similar to previous work, for example, latent diffusion model, and DiffSeg.
>
> Reply: Our model is based on diffusion model but in pixel-space (not latent space). Latent diffusion model (Stable Diffusion) was proposed for image generation task, not segmentation task. We adapt the diffusion model for "ambiguous" medical image segmentation. DiffSeg was proposed for producing a single segmentation mask, whereas we produce a distribution of segmentation masks which reflect expert ambiguity. Also, DiffSeg uses a cross entropy loss while we use KL and MSE loss for our model. Moreover, DiffSeg doesn't use text annotations.  Furthermore, in Table 5 in supplementary, we show ablation on why both text guidance and the ambiguity modeling objective are crucial for performance.
>
>
> 2. The experiment only performs on one testbed (LIDC-IDRI). The generalizability remains a concern.
>
> Reply: To the best of our knowledge, this is the "only" dataset which has both multi-annotator as well as metadata. We used this metadata to curate textual annotations.
>
> 3. Several significant segmentation baselines are missing, for example, nnunet.
>
> Reply: All the methods we compared our model against are SOTA for "ambiguous" medical image segmentation. Other models such as nnUNet are for deterministic segmentations for single annotator setting.
>
> 4. What is the segmentation performance of the other baseline?
>
> Reply: Section 5.1 Table 2 shows our model's performance against 4 baselines. In supplementary, we also show comparison of various preference strategies that we propose and their comparison against baselines.
>
> 5. Using the DDIM for faster sampling is hard to be considered as a novelty.
>
> Reply: We never say that DDIM is the novelty of our paper but rather we show the benefit of incorporating DDIM sampling in order to achieve a speedup, making the approach suitable for large-scale and clinical use.

---

### Meta-Review · Area_Chair_fVTr · 2026-01-08

**Summary:**

Across the reviews, there is consensus that the paper addresses a relevant and timely problem in ambiguous medical image segmentation by combining diffusion models, text conditioning, and preference alignment, but that its overall contribution is incremental relative to existing diffusion and DPO-based frameworks. The rebuttal effectively addresses several concerns, particularly by clarifying the role of Dice-based preference construction, the tradeoff between Dice and GED, and by fixing presentation issues noted by Reviewer 3. However, core weaknesses remain largely unresolved: the novelty is primarily in metric-based preference construction rather than in modeling or optimization principles. The exclusive reliance on a single dataset limits empirical generality, in which all three reviewers consistently pointed out, and author's rebuttal has only stated the fact that other existing datasets are unsuitable; however, it is also a pity that experimental validation is indeed limited. Comparisons to stronger deterministic baselines are absent. Paper presentation and writing can be improved.

Based on the mixed but predominantly lukewarm ratings, limited demonstrated novelty, and constrained empirical scope despite a reasonable rebuttal, this paper is a borderline paper toward rejection (according to the high bar for ICLR), though it shows promise with further validation and sharper positioning of contributions.

**Reviewer Concerns:**

I think many review comments get responses in the rebuttal; the authors have done the best they can in answering questions. However, as pointed out above, the rebuttal is not expected to add new experimental results, so the reliance on a single dataset limitation can not be addressed in the rebuttal.

**Reviewer Scores:**

I think Reviewer 1 SCoE might change score. The authors complaint about Reviewer 1 review quality, however there is no response post rebuttal.

Reviewers 2 has given descent initial scores. Although the rebuttal has responded to asked questions, I believe he/she will not raise scores.

Reviewer 3 questions and suggestions are acknowledged by the author, and the score could be slightly increased.

---

### Decision · Program_Chairs · 2026-01-26

Reject